# Perceived Knowledge, Attitude, and Practices (KAP) and Fear toward COVID-19 among Patients with Diabetes Attending Primary Healthcare Centers in Kuwait

**DOI:** 10.3390/ijerph20032369

**Published:** 2023-01-29

**Authors:** Fatemah M. Alsaleh, Muna Elzain, Zahra K. Alsairafi, Abdallah Y. Naser

**Affiliations:** 1Department of Pharmacy Practice, Faculty of Pharmacy, Kuwait University, Kuwait City 12037, Kuwait; 2Department of Applied Pharmaceutical Sciences and Clinical Pharmacy, Faculty of Pharmacy, Isra University, Amman 11622, Jordan

**Keywords:** KAP, diabetes mellitus, Kuwait, fear, COVID-19

## Abstract

Objectives: To assess perceived fear and to evaluate the level of knowledge, attitude, and prevention practices (KAP) regarding COVID-19 infection among patients with diabetes mellitus (DM) attending primary healthcare centers (PHCs) in Kuwait. This will help evaluate gaps and provide appropriate support to limit the spread of COVID-19 infection in high-risk patients. Methods: A descriptive cross-sectional study was carried out using a self-administered questionnaire. All patients aged 18 years or older attending for follow-up or newly diagnosed with type 1 or type 2 diabetes were eligible to participate in the study. Patients waiting for their regular follow-up appointments at the PHCs were invited verbally to take part in the study. The study excluded patients under the age of 18 and those with significant cognitive or physical impairment that might interfere with independent self-care behavior. The questionnaire included 57 items. The data were analyzed using descriptive statistics. Results: A total of 294 questionnaires were distributed to patients at PHCs in three health districts (Hawally, Capital, and Farwaniya) in Kuwait; 251 patients agreed to participate in the study, yielding a response rate of 85.4%. The study showed that most patients had moderate knowledge (71.1%) of COVID-19. The majority of correctly responded questions were about the mode of COVID-19 transmission, the most common clinical presentations, and at-risk people. On the other hand, 83.7% (n = 210) identified common cold symptoms (stuffy nose, runny nose, and sneezing) as COVID-19 symptoms. More than half of the patients (n = 146, 58.2%) were unable to identify uncommon COVID-19 symptoms, such as diarrhea and skin rash or discoloration. Most patients had a positive attitude (90.9%) and good prevention practices (83.6%). The overall fear score of the participating patients was 21.6 ± 6.5 (61.7%). Conclusions: Regardless of the positive attitude and good preventive practices of the patients, they had moderate knowledge levels about COVID-19. This indicates that there are significant knowledge gaps that still need to be filled. Different strategies can be used for this purpose, such as social media and public information campaigns. Supporting psychological well-being is vital for at-risk patients during a pandemic.

## 1. Introduction

The coronavirus disease 2019 (COVID-19) pandemic has emerged as one of the greatest challenges threatening humankind in the world. It is a respiratory infectious disease caused by the severe acute respiratory syndrome coronavirus 2 (SARS-CoV-2) [1]. The first case occurred in Wuhan, China, at the end of 2019 [2], after which the infectious disease spread around the world, quickly reaching 100,000 cases. By the end of January 2020, the World Health Organization (WHO) declared COVID-19 a global health emergency and called for a worldwide collaborative effort to defeat the spread [2]. To date, there have been more than 500 million confirmed cases of COVID-19, including more than 6 million deaths [3].

COVID-19 is characterized by rapid transmission through diverse routes, including respiratory droplets from an infected person, aerosols (airborne transmission), possibly in a healthcare environment, and direct contact with infected surfaces (e.g., doorknobs) [4,5,6]. Patients with COVID-19 can be asymptomatic or experience a variety of symptoms ranging from mild to severe, or even life-threatening. Elderly people and patients with pre-existing medical conditions, such as diabetes mellitus (DM) and cardiovascular, renal, and respiratory diseases, are considered the highest at-risk groups for COVID-19 infection, with elevated severity and higher mortality rates [7,8,9,10,11]. In a large observational study conducted in China, 173 patients (out of 1099) had severe COVID-19, of which 16.2% had DM [12]. In another study that recruited a sample of 52 patients with severe COVID-19, patients with DM constituted 22% of the 32 in the group who did not survive due to COVID-19 complications [13]. 

In Kuwait, there have been more than 600,000 confirmed COVID-19 cases, with 2555 cumulative deaths since the start of the pandemic [3]. DM is one of the most common chronic diseases, with a prevalence of 13.5% among the adult population [14]. As part of government efforts to limit the spread of the disease, the Ministry of Health (MOH) imposed preventive strategies against COVID-19, consistent with the WHO, such as isolation of infected and suspected individuals, social distancing, wearing of face masks and gloves, use of proper hygiene measures, and administration of vaccines [15]. Currently, in the absence of an effective cure against COVID-19, high-risk individuals, such as those with DM, must stringently follow precautionary measures against COVID-19. 

Effective pandemic control can be attained through good knowledge, attitude, and practices (KAP) toward COVID-19, in line with the “KAP” theory. According to the KAP theory, human health behavioral change is achieved through the acquisition of the right knowledge, generation of attitudes, and adoption of behaviors (or practices) in three successive processes [16]. In this context, many studies have shown that the KAP level of patients was associated with either effective or poor prevention and/or management of their illnesses [17,18,19,20]. The rapid increase in cases, deaths, and uncertainty about COVID-19 around the world has been claimed to increase psychosocial distress among the general population [21]. More attention should be directed toward people with chronic diseases, such as DM, who often have concomitant psychological disturbances, such as depression and anxiety [22]. Many systematic reviews and meta-analyses have been conducted to identify studies assessing KAP toward COVID-19 across different countries [23,24,25] and populations [26,27]. To the best of our knowledge, there is scarce data assessing KAP toward COVID-19 among high-risk patients [28,29,30,31] and assessing their perceived level of fear [32,33,34]. Thus, this study was undertaken to assess KAP and the perception of fear toward COVID-19 in patients with DM attending primary healthcare centers (PHCs) in Kuwait. In the following sections, the methods used to conduct the study, the main results, and the discussion will be provided in detail.

## 2. Methods

### 2.1. Study Design and Setting

A quantitative, prospective, cross-sectional study using a self-administered questionnaire was undertaken from April to May 2021. This study was conducted in PHCs delivering specialized diabetes healthcare services, which are distributed over the five health districts in Kuwait (Al Jahra, Capital, Hawally, Al Farwaniya, and Al Ahmadi). 

### 2.2. Study Participants and Data Collection Procedures

This study included patients aged 18 years or older on a follow-up or with a new diagnosis of type 1 or type 2 DM who were willing to sign a consent form. Patients younger than 18 years and those with a clear cognitive or physical disability that could interfere with independent self-care behavior were excluded. Random selection was initially made at the level of the polyclinic to select sites for data collection; then, a proportional number of patients was recruited from each PHC.

### 2.3. Study Instrument 

The study questionnaire was designed based on an extensive literature review of previously published studies [28,30,35,36,37] and according to the WHO and MOH COVID-19 guidelines for the public [38,39]. Two senior researchers reviewed the questionnaire for clarity and relevance. It comprised five sections. The first section included 28 items and had possible responses of “true”, “false”, or “I don’t know” to assess patients’ knowledge of COVID-19 in terms of methods of transmission, symptoms, treatments, high-risk groups, prevention, and spread control strategies. An “incorrect” or “I don’t know” answer was given 0 points, while 1 point was awarded for a correct answer, giving a possible score ranging from 0 to 28. Using the measures and cut-off points from other studies [30,35,36], the patients’ overall knowledge was categorized as good, moderate, or poor if the total score was 80–100%, 60–79%, or less than 60%, respectively.

The second section comprised seven items to evaluate the patients’ attitudes toward COVID-19. The response to each item was indicated on a five-point Likert scale as follows: strongly agree (5 points), agree (4 points), neutral (3 points), disagree (2 points), and strongly disagree (1 point); thus, the total possible score ranged from 7 to 35. Using the measures and cut-off points from other studies [30,35,36], overall attitude was classified as positive, neutral, or negative if the total score was 80–100%, 60–79%, or less than 60%, respectively. 

The third section explored patients’ practices and included 15 questions related to the prevention practices recommended by the WHO and MOH guidelines. The practices questions were scored using a three-point Likert scale, with a total score ranging from 15 to 45. For all questions, participants were asked to state the frequency with which they followed preventative practices by answering always/often (3 points), sometimes (2 points), or rarely/never (1 point) for questions 1 to 13 and question 15. The scoring for question 14 was as follows: always/often (1 point), sometimes (2 points), or rarely/never (3 points). Using the measures and cut-off points from other studies [30,35,36], the participants’ overall practices were categorized as good, fair, or poor if the total score was 80–100%, 60–79%, or less than 60%, respectively.

The fourth section of the questionnaire assessed the intensity of fear using the Fear of COVID-19 Scale (FCV-19S). The FCV-19S is a validated instrument developed by Ahorsu et al. [40] that consists of seven questions with a five-point Likert scale. For all questions, participants were asked to state their degree of agreement as strongly agree (5 points), agree (4 points), neutral (3 points), disagree (2 points), and strongly disagree (1 point) for a total possible score ranging from 7 to 35 points. Higher scores indicate higher levels of perceived fear. 

The last part of the questionnaire included questions on patients’ demographics (nine items) and their medical information (seven items). An Arabic version of the questionnaire was prepared and validated using a back-translation method.

### 2.4. Sample Size 

According to statistics obtained from the MOH during the initial fieldwork, the total number of visits to specialized diabetes PHCs (n = 87) in 2020 was 944,844. The sample size was calculated using the online Raosoft^®^ calculator based on the assumption that the proportion of responses to most of the main questions would be 50. Using a margin of error of 5% and a confidence interval (CI) of 95%, the minimum sample size was determined to be 384. Assuming a response rate of 80%, a larger sample size of 480 was considered. Table 1 shows the total number of patients in each PHC.

### 2.5. Piloting Phase

A pilot study was initially performed to evaluate the clarity of the questionnaire and the feasibility of the study procedures. For this purpose, 15 questionnaires were distributed over three PHCs selected randomly from different governorates. The pilot study ended with no modifications.

### 2.6. Data Analysis

Statistical analysis was performed using SPSS version 27 (IBM Corp., Armonk, NY, USA). Data from closed-ended questions were initially coded and entered into SPSS. Descriptive statistics were presented using frequencies and percentages for categorical variables and as means (standard deviation; SD) for normally distributed continuous variables or median (interquartile range; IQR) for non-normally distributed data. Binary logistic regression was used to identify predictors of patients’ perceived fear and knowledge, attitude, and prevention practices toward COVID-19. The mean participants’ scores concerning their fear and knowledge, attitude, and prevention practices toward COVID-19 were used as the cut-off point to define the dummy variables for the logistic regression. A confidence interval (CI) of 95% (*p* < 0.05) was applied to represent the statistical significance of the results, and the level of significance was assigned as 5%.

## 3. Results 

### 3.1. Patients’ Demographics and Medical Information

A total of 294 questionnaires were distributed to patients at PHCs in Kuwait that were chosen randomly from three health districts: Hawally (n = 77), Capital (n = 67), and Farwaniya (n = 107). Two hundred and fifty-one agreed to participate in the study, yielding a response rate of 85.4%. 

The median age of the patients in this study was 52 years, with a range of 18 to 87 years old. More than half of the study sample were males (n = 166, 66.1%), Kuwaitis (n = 135, 53.8%), married (n = 176, 70.1%), and with a minimum degree of diploma (diploma (n = 58, 23.1%); bachelor’s degree (n = 84, 33.5%); postgraduate degree (n = 13, 5.2%)). Most patients were employed either in the governmental or private sectors (n = 142, 56.6%), with more than one-third having a monthly income between 500 and 1000 Kuwaiti dinars (KWD) (n = 93, 37.1%) (Table 2).

Most of the study participants had type 2 diabetes (n = 227, 90.4%) and were using oral hypoglycemic agents (n = 168, 67%) (Table 3). However, only 22 (8.8%) patients had their HbA1c under 7%, which is the target recommended by the American Diabetes Association [41]. More than half of the study sample had concomitant cardiovascular diseases, such as hypertension and heart disease. Only 15.2% of the patients (n = 38) had a previous COVID-19 infection. At the time of data collection, most of the study participants had not received the COVID-19 vaccine (n = 149, 59.4%).

### 3.2. Patients’ Knowledge about COVID-19

With a maximum score of 28, the overall mean knowledge score was 19.9 ± 3.4 (which is equivalent to 71.1% of the total attainable score). Most of the study sample had an overall “moderate” knowledge level (n = 154, 61.4%), whereby they achieved scores between 60 and 79%. Only 19.9% (n = 50) had “good” knowledge, and 16.3% (n = 41) had “poor” knowledge. The majority of patients recognized the different routes of COVID-19 transmissions, such as respiratory droplets (n = 244, 97.2%) and contact with contaminated surfaces (n = 238, 94.8%). In contrast, the majority did not have the knowledge that eating or being in contact with wild animals (n = 139, 55.4%) and using insulin injections (n = 155, 61.8%) do not contribute to the transmission of COVID-19. 

In terms of knowledge about the clinical presentation, most patients (n = 233, 92.8%) correctly identified fatigue, fever, dry cough, muscle aches, and shortness of breath as the main symptoms of COVID-19; however, 83.7% (n = 210) of patients identified symptoms of the common cold (stuffy nose, runny nose, sneezing) as COVID-19 symptoms. In addition, most patients in the study (n = 146, 58.2%) could not identify other uncommon symptoms of COVID-19, such as diarrhea and skin discoloration. Most patients were able to identify high-risk group patients who are more prone to COVID-19 infection or more likely to have a more severe infection, such as patients with diabetes (n = 192, 76.5%) and the elderly (n = 211, 84.1%) (Table 4). 

As for knowledge of the management of COVID-19, most patients (n = 198, 78.9%) were aware that at the time of the study there was no effective treatment but that early supportive measures could help in the recovery process. In contrast, 53% of the patients (n = 133) believed that antibiotics were effective for COVID-19 management. The majority of patients (n = 147, 58.6%) did not know that there was an effective vaccine to prevent COVID-19 infection. Regarding knowledge about the prevention and control of COVID-19, most patients believed that proper hand washing (n = 236, 94%), wearing face masks (n = 236, 94%), avoiding crowded places (n = 206, 82.1), and isolating infected patients (n = 241, 96%) can prevent the spread of COVID-19 (Table 5). 

### 3.3. Patients’ Attitudes toward COVID-19

The overall prevalence of positive attitudes toward COVID-19 among patients with diabetes attending PHCs was 90.4% (n = 227). The mean score was 31.8 ± 3.4 (which is equivalent to 90.9% of the total attainable score). Most patients strongly agreed that undertaking preventive measures is effective to prevent or slow COVID-19 infection. In this context, 204 patients (81.3%) strongly agreed that regular hand washing, maintaining social distancing, and covering the face with masks are necessary measures to protect oneself from COVID-19. Almost three-quarters of the patients strongly agreed that they should stay at home if they were feeling sick (n = 188, 74.9%), and when infected, they would accept being isolated in a health facility (n = 192, 76.5%). The majority of patients (n = 170, 67.7%) trusted the prevention measures recommended by the MOH in Kuwait and believed that the disease would eventually be controlled with time (n = 131, 52.2%) (Table 6). 

### 3.4. Patients’ Practices of Prevention from COVID-19 Infection

With a maximum score of 45, the overall mean practices score was 37.6 ± 3.4 (which is equivalent to 83.6% of the total attainable score). Over two-thirds (n = 181, 72.1%) of the patients reported good prevention practices. Most patients always or often wore face masks (n = 229, 91.2%) and maintained social distancing (n = 194, 77.3%) when leaving their homes. Similarly, most of them stayed at home when they felt sick (n = 201, 80.1%) and avoided large gatherings (n = 192, 76.5%) (Table 7).

### 3.5. Patients’ Perceptions of Fear with Regard to COVID-19

The study sample had an overall mean fear score of 21.6 ± 6.5 (which is equivalent to 61.7% of the total attainable score). Two-thirds (n = 167, 66.5%) of the patients either strongly agreed or agreed that they were afraid of acquiring COVID-19 infection, and 65.3% (n = 164) were even uncomfortable thinking about COVID-19. Similarly, more than half of the patients were afraid of losing their lives and became anxious or nervous upon watching COVID-19 news, as reported by 54.2% (n = 136) and 52.6% (n = 132), respectively (Figure 1). 

### 3.6. Factors Associated with Patients’ Perceived Fear and Knowledge, Attitude, and Prevention Practices toward COVID-19

Binary logistic regression analysis determined that divorced patients and those with postgraduate education were less likely to have perceived fear of COVID-19 (*p* ≤ 0.05). Furthermore, unemployed and non-Kuwaiti patients were more likely to be knowledgeable about COVID-19 and have better prevention practices, respectively, compared to others. Conversely, there was no statistically significant difference between patients in terms of their attitudes toward COVID-19 (*p* ≥ 0.05). Non-Kuwaiti patients showed significantly better prevention practices compared to Kuwaitis (Table 8).

## 4. Discussion 

To the best of our knowledge, this is the first study to evaluate fear of COVID-19 among patients with DM attending PHCs in Kuwait and to assess their KAP level. The study found that most of the patients had a moderate level of knowledge (61.2%). This is compatible with a study by Naser et al. [42] that was conducted on the public in several Middle Eastern countries, including Kuwait, which found that two-thirds of the participants (66.1%) had moderate knowledge about COVID-19. Consistent with our study, a study conducted in Syria [43] revealed that 60% of the population had moderate knowledge of COVID-19. In contrast, studies performed in this context in China [37,44,45] reported that over 80% of participants had good knowledge of COVID-19. This difference in knowledge levels between participants in China and those in Kuwait or other Middle Eastern countries may be because China is the country where COVID-19 originated, so its population might have had better knowledge than other populations worldwide.

Most patients in the current study knew the common transmission modes of COVID-19, such as respiratory droplets and touching contaminated surfaces; however, only a minority knew that the infection does not spread through insulin injections. This can be explained by the fact that the risk of infection spread through injections and blood contamination was not explained in the guidelines provided by the WHO [34] and MOH in Kuwait [39]. In contrast, the National Institute of Environmental Health Sciences (NIEHS) [46] mentioned that COVID-19 is not a bloodborne infection and thereby does not spread via needlesticks. Similar to our results, in a study conducted in India [28], most patients with type 1 DM identified the common transmission methods of COVID-19, such as respiratory droplets (94.8%) and touching contaminated surfaces and then their mouth, nose, or eyes (95.2%); however, the majority (60%) did not know that the infection does not spread through insulin injections. In terms of COVID-19 symptoms, regardless of the fact that the majority of patients in the current study were able to identify the main symptoms of COVID-19, most failed to distinguish the symptoms of COVID-19 from those of the common cold. In comparison, the majority of patients (68.7%) in the study conducted by Naser et al. were able to differentiate between different symptoms [42]. This may be due to the different targeted populations, as the current study was conducted on patients with DM, while the target population in Naser et al.’s study [42] was the public. In addition, their study included participants from other Middle Eastern countries. 

Regarding knowledge questions on the prevention and control of COVID-19, most patients answered this section correctly and knew that hand washing, wearing face masks, avoiding crowded places, and isolating infected patients are effective preventive measures. These results are comparable with the studies by Naser et al. [42] and Addis et al. [31] in which the participants showed good knowledge regarding prevention practices for COVID-19. The current study revealed that 85.3% of the patients did not know that some diabetes medications should be stopped during COVID-19 infection, which can be alarming. Fortunately, most of the study sample (84.9%) had never experienced a previous COVID-19 infection.

Regarding attitudes, most patients showed a positive attitude, which is consistent with other studies [44,45,47,48]. Most of the patients reported that they trusted the preventative measures implemented by the MOH, and more than half strongly agreed that the pandemic would eventually be brought under control; these findings are similar to the results of a study that was conducted in Bosnia and Herzegovina [35,49]. This positive attitude and trust in the MOH were reflected in good practices and following the preventive protocols established by the government.

The results from the current study showed that 72.1% of the patients had overall good prevention practices. This was reflected in avoiding large gatherings (76.5%), applying social distancing measures (77.3%), and wearing face masks (91.2%). In contrast, a quarter (25.1%) of the patients always/often wore gloves when leaving home, which was comparable to a study performed in the KSA [50] in which 36.4% of the participants wore gloves when going out. In the current study, only a minority of patients (21.1%) used herbs or traditional medicine to prevent the spread of COVID-19. This is compatible with the protocols set by the MOH of Kuwait [39], which recommends against the use of herbs and traditional medicine to prevent COVID-19. In contrast, in a study conducted in Ethiopia [30] on patients with hypertension and/or diabetes, the results showed that 64.3% of the patients had used herbs and/or traditional medicines to protect themselves or prevent COVID-19 infection. 

As for the patients’ perceptions of fear with regard to COVID-19, the current study showed a mean fear score of 21.6 ± 6. Although most patients were afraid of COVID-19 in general, the majority did not experience clammy hands (53.0%), insomnia (62.5%), or palpitations (57.8%). The score is comparable with a study conducted in Canada [34] in which the mean score was 17.05 ± 4.38, and another study in the KSA [51], which found a mean score of 17.3 ± 5.21. Consistent with the current study, the participants in these two studies had higher mean scores on questions related to being afraid of COVID-19 in general and lower scores related to insomnia, anxiety, and palpitations. Unsurprisingly, this study showed that people with a postgraduate level of education had less fear of COVID-19 compared to others. This could be because their level of education positively impacted their psychological well-being [52,53]. Similar to the results from a study by Doshi et al. in 2021 [54], a significant association was shown between fear and marital status (i.e., divorced patients had less fear compared to others). This could be attributed to the fact that married people have a greater sense of responsibility and financial commitment toward their loved ones compared to others, and the pandemic has affected job security worldwide. Unemployed patients in this study had more knowledge about COVID-19 compared to others. This could be because they have more free time than those who are working; thus, they have greater knowledge about pandemics with better coping skills for emergencies compared to others [55]. Non-Kuwaiti patients showed significantly better prevention practices compared to Kuwaitis. This could be explained by the fact that healthcare delivery is supported by the government and is free of charge for citizens. Conversely, non-Kuwaitis must pay for their medical expenses, which may have affected their adherence to preventive practices during the COVID-19 pandemic.

### Study Strengths and Limitations 

This is the first study to investigate the KAP and fear of COVID-19 among patients with DM, which is a common high-risk group for more severe infectious complications. The research was conducted across three governorates and achieved an adequate response rate of 85.4%, which may enhance the generalizability of the findings. Randomizing the PHCs for selecting sites for data collection is another strength. Finally, the study tool was designed, piloted, and assessed based on content and face validity.

Insufficient time for data collection was the main limitation precluding data collection from the five governorates, as initially planned. Another limitation is that the COVID-19 pandemic affected the number of PHCs that were open to deliver their services. Moreover, the number of patients that could be seen in each PHC decreased as part of the precautionary measures imposed by the government. 

This study was conducted during a time when the COVID-19 pandemic was in an active phase; therefore, the research team undertook the following precautionary measures to optimize safety: the data collector wore two face masks with frequent hand disinfection, the patients were offered disposable gloves and hand sanitizers, and the questionnaires were enveloped separately before and after being received from the patients. Regardless, many patients were still hesitant to fill out the questionnaire. 

## 5. Conclusions

In conclusion, this study showed that the current knowledge level about COVID-19 among patients with DM was moderate. They showed a positive attitude and good preventive practices. Fortunately, most patients had not experienced a prior COVID-19 infection. The patients expressed poor knowledge with regard to the changes that must be introduced to their treatment regimen if they become infected. The COVID-19 outbreak added additional psychological distress to this high-risk population, which could exacerbate any underlying psychological disorders. As most of the participants showed moderate knowledge of COVID-19, attention must be paid to those who showed low knowledge levels, as they were more likely to have poor attitudes and prevention behavior. The MOH and public health authorities should propose future policies and interventions that use a “patient-centered” approach rather than a “disease-centered” approach to identify vulnerable populations and prioritize policies and communication efforts to improve patients’ knowledge and behavior. 

### Implications for the Future and Practice 

The findings of this study highlight the perceived fear and KAP levels about COVID-19 among a high-risk group (i.e., patients with DM). Although the majority of patients showed a moderate level of knowledge about COVID-19, there was still a proportion with low knowledge levels. In addition, patients with lower education levels had more perceived fear of COVID-19. These results should raise the awareness of policymakers to propose targeted interventions that ensure the psychological well-being of targeted patients during the pandemic period. Although the MOH was highly effective in managing manpower and providing medical supplies during the outbreak, emotional support for patients is crucial. The MOH should provide patients with psychological assistance, such as a hotline service with psychiatrists, to provide them with stress relief activities. Future qualitative studies are required in this field to help further explore and address the needs of patients with DM during pandemics. Therefore, to alleviate the psychological and knowledge impairment of targeted groups of patients, PHCs are encouraged to conduct educational campaigns targeting patients to improve their knowledge and awareness about COVID-19 and reassure them about the effectiveness of the applied prevention measures in providing safe conditions. 

## Figures and Tables

**Figure 1 ijerph-20-02369-f001:**
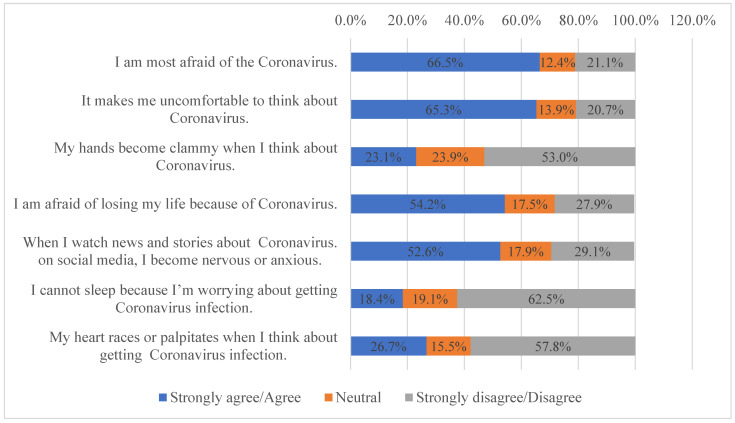
Patient’s perceptions of fear represented by percentages of those who strongly agree/ agree, were neutral, or strongly disagree/disagree with each statement.

**Table 1 ijerph-20-02369-t001:** Questionnaire distribution plan across the governates, Kuwait.

Governorate	Number of Diabetes Clinics Visits in 2020	Percentage	Number of Questionnaires per Governorate
Hawally	166,739	18%	86
Capital	139,173	15%	72
Al Ahmadi	244,941	26%	125
Farwaniya	264,524	28%	135
Al Jahra	129,367	13%	62
Total	944,844	100%	480

**Table 2 ijerph-20-02369-t002:** Patients’ demographics (n = 251).

Characteristics	n (%)
Age	Median 52 years (min. 18, max. 87)
18 to 34 years 35 to 49 years 50 to 64 years 65 years and older	20 (8.0) 71 (28.3) 138 (55.0) 22 (8.8)
Gender	
Male Female	166 (66.1) 85 (33.9)
Marital status	
Single Married Divorced Widowed	26 (10.4) 176 (70.1) 26 (10.4) 23 (9.2)
Nationality	
Kuwaiti Non-Kuwaiti	135 (53.8) 116 (46.2)
Educational status	
Illiterate High school Diploma Bachelor’s degree Postgraduate degree	7 (2.8) 89 (35.5) 58 (23.1) 84 (33.5) 13 (5.2)
Employment status	
Student Employed (government/private) Self-employed Unemployed Retired	7 (2.8) 142 (56.6) 17 (6.8) 26 (10.4) 59 (23.5)
Monthly income	
Less than 500 KWD 500 to 1000 KWD 1000 to 2000 KWD More than 2000 KWD	76 (30.3) 93 (37.1) 57 (22.7) 24 (9.6)

KWD: Kuwaiti Dinar.

**Table 3 ijerph-20-02369-t003:** Patients’ medical information (n = 251).

Characteristics	n (%)
Type of diabetes	
Type 1 diabetes Type 2 diabetes	24 (9.6) 227 (90.4)
Duration of diabetes	Median 7 years (min. 0.2, max. 41)
Less than 1 year 1 to less than 5 years 5 to less than 10 years 10 years or more	7 (2.8) 56 (22.3) 82 (32.7) 106 (42.2)
Last HbA1c value	Median 8% (min. 5.8%, max. 13.0%)
Less than 7% 7% and above	22 (8.8) 229 (91.2)
Treatment	
Diet and exercise only Oral hypoglycemic agents Insulin Combined oral hypoglycemic agents + insulin	6 (2.4) 168 (67.0) 31(12.3) 46 (18.3)
Other chronic conditions *	
Hypertension Heart disease Cancer Respiratory disease (e.g., COPD, asthma) Autoimmune diseases (e.g., thyroid diseases, rheumatoid arthritis)	120 (47.8) 40 (15.9) 0 (0.0) 25 (10.0) 14 (5.6)
Past diagnosis of COVID-19	
Yes ** No	38 (15.1) 213 (84.9)
Receipt of COVID-19 vaccine ****	
Yes *** No	101 (40.2) 149(59.4)

KWD: Kuwaiti Dinar, HbA1c: Glycosylated hemoglobin type 1AC. * These conditions occurred either alone or in combination. ** Any changes introduced to their diabetes treatment regimen: no changes (n = 31, 81.6%); yes, stopping oral hypoglycemic agents and adding insulin during the infection period (n = 7, 18.4%). *** Brand of vaccine received: Pfizer (n = 62, 61.4%), AstraZeneca (n = 39, 38.6%). **** Data were missing from one patient.

**Table 4 ijerph-20-02369-t004:** Patients’ responses to questions related to knowledge about COVID-19 in terms of transmission, clinical presentation, and high-risk groups (n = 251).

Knowledge Questions	Correct n (%)	Incorrect n (%)
Coronavirus infection spreads from person-to-person within close distance of each other (about 6 feet; 2 m).	166 (66.1)	85 (33.9)
Coronavirus infection spreads through respiratory droplets, which occur when infected people cough and sneeze.	244 (97.2)	7 (2.8)
The infection can spread by touching a surface or object, to which the virus is attached, and then touching one’s mouth, nose, or, perhaps, eyes.	238 (94.8)	13 (5.2)
Eating wild animals or direct contact with them can lead to infection with the Coronavirus.	112 (44.6)	139 (55.4)
The coronavirus does not spread through the use of insulin injection. *	95 (37.8)	155 (61.8)
Home deliveries can contribute to the spread of Coronavirus infection.	203 (80.9)	48 (19.1)
Infected people cannot transmit the Coronavirus to others if they have no fever.	172 (68.5)	79 (31.5)
The main symptoms of Coronavirus infection are fever, fatigue, dry cough, muscle aches, and shortness of breath.	233 (92.8)	18 (7.2)
A stuffy nose, runny nose, and sneezing are the most common symptoms of Coronavirus infection.	41 (16.3)	210 (83.7)
Diarrhea, skin rash, or discoloration are symptoms of Coronavirus infection.	105 (41.8)	146 (58.2)
All people infected with Coronavirus develop symptoms and feel unwell.	180 (71.7)	71 (28.3)
All people infected with Coronavirus develop severe and serious symptoms. **	202 (80.5)	47 (18.7)
People with diabetes mellitus are more likely to develop severe symptoms and even die from Coronavirus infection. *	192 (76.5)	58 (23.1)
Coronavirus infection is more dangerous in the elderly than in the young.	211 (84.1)	40 (15.9)
It is not necessary for children and young people to take the precautions required to limit the spread of Coronavirus infection. *	206 (82.1)	44 (17.5)

* Data were missing from one patient. ** Data were missing from two patients.

**Table 5 ijerph-20-02369-t005:** Patients’ responses to questions related to knowledge about COVID-19 management, prevention, and control (n = 251).

Knowledge Questions	Correct n (%)	Incorrect n (%)
Antibiotics are an effective treatment for Coronavirus infection. *	117 (46.6)	133 (53.0)
At present, there is no effective treatment for Coronavirus infection, but early symptomatic and supportive treatment can help most patients recover from the disease.	198 (78.9)	53 (21.1)
At present, there is no effective vaccine to prevent Coronavirus infection.	104 (41.4)	147 (58.6)
After coughing, sneezing, or nose-blowing in a public place, wash hands with soap and water for at least 20 s, or use a hand sanitizer that contains at least 60% alcohol.	236 (94.0)	15 (6.0)
People can wear general medical masks to prevent infection with Coronavirus.	236 (94.0)	15 (6.0)
People should only wear a mask if they are infected with the Coronavirus, or if they are caring for a person suspected of having the infection.	183 (72.9)	68 (27.1)
Wearing gloves is more effective than regular hand washing with soap and water.	177 (70.5)	74 (29.5)
Avoiding crowded places and maintaining a minimum distance of 1 meter from others can prevent the spread of Coronavirus.	206 (82.1)	45 (17.9)
Outdoor gatherings are safer than indoor gatherings.	201 (80.1)	50 (19.9)
Isolation and symptomatic treatment of patients with Coronavirus infection are effective ways to reduce the spread of the disease.	241 (96.0)	10 (4.0)
If you have accidentally come in contact with a person infected with Coronavirus, you must immediately isolate yourself for a minimum period of 14 days, and you must go immediately to the nearest hospital if you feel unwell.	234 (93.2)	17 (6.8)
Healthy food and drinking water help to increase the body’s immunity and resistance to getting Coronavirus infection.	227 (90.4)	24 (9.6)
Some diabetes medications should be stopped in patients with Coronavirus infection.	37 (14.7)	214 (85.3)

* Data were missing from one patient.

**Table 6 ijerph-20-02369-t006:** Patients’ responses to questions related to attitudes toward COVID-19 (n = 251).

Attitude Questions	Strongly Agree n (%)	Agree n (%)	Neutral n (%)	Disagree n (%)	Strongly Disagree n (%)
Regular hand washing, maintaining social distancing, and covering the face with masks are necessary to protect from COVID-19.	204 (81.3)	39 (15.5)	4 (1.6)	2 (0.8)	2 (0.8)
To protect me from Coronavirus exposure, I should stay at home if I’m sick unless seeking medical care.	188 (74.9)	46 (18.3)	9 (3.6)	8 (3.20)	0 (0.0)
I will perform a coronavirus test if I suspect symptoms of Coronavirus infection.	185 (73.7)	54 (21.5)	6 (2.4)	6 (2.4)	0 (0.0)
If I got infected with Coronavirus, I would accept isolation in a health facility if necessary.	192 (76.5)	47 (18.7)	5 (2.0)	6 (2.4)	1 (0.4)
I trust the prevention measures recommended by the MOH	170 (67.7)	51 (20.3)	15 (6.0)	11 (4.4)	4 (1.6)
I believe that vaccines can reduce the spread of Coronavirus infection.	134 (53.4)	79 (31.5)	30 (12.0)	7 (2.8)	1 (0.4)
COVID-19 will eventually be brought under control.	131 (52.2)	91 (36.3)	25 (10.0)	3 (1.2)	1 (0.4)

MOH: Ministry of Health, Kuwait.

**Table 7 ijerph-20-02369-t007:** Patients’ responses to questions related to practices of prevention from COVID-19 (n = 251).

Practice Questions	Always/Often n (%)	Sometimes n (%)	Rarely/Never n (%)
When you leave home, do you wear a mask? *	229 (91.2)	18 (7.2)	3 (1.2)
When you leave home, do you wear gloves?	63 (25.1)	106 (42.2)	82 (32.7)
When you leave home, do you maintain a distance of at least 1 meter from other people?	178 (70.9)	68 (27.1)	5 (2.0)
When you leave home, do you apply social distancing measures and avoid hand shaking or kissing?	194 (77.3)	51 (20.3)	6 (2.4)
Do you wash your hands using soap for at least 20 s?	178 (70.9)	70 (27.9)	3 (1.2)
Do you avoid touching your nose, eyes, and mouth with your hands before washing them?	138 (55.0)	110 (43.8)	3 (1.2)
Do you cover your mouth and nose after coughing/sneezing and wash your hands thereafter? *	169 (67.3)	72 (28.7)	9 (3.6)
Do you clean/disinfect frequently touched objects and surfaces?	140 (55.8)	74 (29.5)	37 (14.7)
Do you avoid close contact with people who have flu/cold symptoms such as coughing, sneezing, or fever?	222 (88.4)	20 (8.0)	9 (3.6)
Do you stay home when you feel sick unless you get medical care?	201 (80.1)	45 (17.9)	5 (2.0)
Do you avoid large gatherings?	192 (76.5)	51 (20.3)	8 (3.2)
Do you avoid consuming food outside the home to prevent the spread of Coronavirus infection?	155 (61.8)	67 (26.7)	29 (11.6)
Do you practice sports more regularly than before the Coronavirus?	46 (18.3)	85 (33.9)	120 (47.8)
To limit the spread of Coronavirus, do you use herbs or traditional medicines?	53 (21.1)	101 (40.2)	97 (38.6)
To avoid catching Coronavirus infection, do you eat fruits and vegetables?	121 (48.2)	87 (34.7)	43 (17.1)

* Data were missing from one patient.

**Table 8 ijerph-20-02369-t008:** Predictors of patients’ perceived fear and knowledge, attitude, and prevention practices toward COVID-19.

Characteristics	Odds Ratio of Fear (95% Confidence Interval)	Odds Ratio of Having Better Knowledge (95% Confidence Interval)	Odds Ratio of Having a Better Attitude (95% Confidence Interval)	Odds Ratio of Having Better Prevention Practices (95% Confidence Interval)
Age				
18 to 34 years (reference group) 35 to 49 years 50 to 64 years 65 years and older	1.00	1.00	1.00	1.00
	1.05 (0.61–1.83)	1.01 (0.58–1.78)	1.55 (0.85–2.82)	1.59 (0.92–2.77)
	0.85 (0.51–1.39)	1.06 (0.63–1.76)	0.83 (0.49–1.40)	0.98 (0.60–1.61)
	1.34 (0.55–3.25)	1.36 (0.53–3.47)	1.49 (0.56–3.95)	0.51 (0.20–1.28)
Gender				
Male (reference group) Female	1.00	1.00	1.00	1.00
	0.89 (0.53–1.50)	0.96 (0.56–1.65)	1.07 (0.62–1.85)	1.10 (0.65–1.86)
Marital status				
Single (reference group) Married Divorced Widow	1.00	1.00	1.00	1.00
	1.52 (0.88–2.61)	0.87 (0.50–1.53)	1.60 (0.91–2.78)	1.16 (0.68–2.00)
	0.36 (0.15–0.87) *	0.99 (0.43–2.28)	0.50 (0.22–1.13)	0.47 (0.20–1.13)
	1.45 (0.60–3.49)	1.46 (0.58–3.70)	0.83 (0.34–1.99)	1.06 (0.45–2.49)
Nationality				
Kuwaiti (reference group) Non-Kuwaiti	1.00	1.00	1.00	1.00
	1.00 (0.61–1.64)	1.55 (0.92–2.59)	0.13 (1.50–0.88)	2.49 (1.49–4.14) **
Educational status				
Illiterate (reference group) High school Diploma Bachelor’s degree Postgraduate degree	1.00	1.00	1.00	1.00
	1.01 (0.60–1.70)	0.93 (0.55–1.59)	0.81 (0.47–1.39)	1.28 (0.76–2.15)
	1.15 (0.64–2.06)	0.70 (0.39–1.27)	0.94 (0.51–1.73)	0.91 (0.51–1.64)
	1.14 (0.67–1.93)	1.18 (0.68–2.03)	1.22 (0.70–2.12)	0.69 (0.41–1.17)
	0.07 (0.01–0.53) *	1.42 (0.42–4.74)	1.23 (0.37–4.11)	2.71 (0.81–9.04)
Employment status				
Student (reference group) Employed (government/private) Self-employed Unemployed Retired	1.00	1.00	1.00	1.00
	1.02 (0.62–1.68)	0.78 (0.46–1.30)	1.22 (0.72–2.05)	1.56 (0.94–2.59)
	1.02 (0.38–2.72)	1.15 (0.41–3.21)	0.76 (0.28–2.06)	1.02 (0.38–2.73)
	0.89 (0.40–2.01)	3.81 (1.27–11.41) *	0.60 (0.26–1.35)	0.98 (0.43–2.21)
	1.19 (0.66–2.14)	0.80 (0.44–1.45)	1.31 (0.70–2.45)	0.67 (0.37–1.21)
Monthly income				
Less than 500 KWD (reference group) 500 to 1000 KWD 1000 to 2000 KWD More than 2000 KWD	1.00	1.00	1.00	1.00
	0.71 (0.43–1.19)	1.04 (0.62–1.77)	0.72 (0.42–1.23)	0.85 (0.51–-1.42)
	1.20 (0.66–2.18)	1.08 (0.59–1.99)	0.82 (0.45–1.51)	0.72 (0.40–1.31)
	0.42 (0.17–1.01)	1.27 (0.52–3.08)	1.70 (0.65–4.44)	0.97 (0.42–2.25)

KWD: Kuwaiti Dinar. * *p*-value < 0.05. ** *p*-value < 0.01.

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
