# Peer review of "Perceived Knowledge, Attitude, and Practices (KAP) and Fear toward COVID-19 among Patients with Diabetes Attending Primary Healthcare Centers in Kuwait"

_ijerph, 2023, doi:10.3390/ijerph20032369_

Round 1

Reviewer 1 Report

Overall, this manuscript is well-written. Congratulations!

However, there are areas for improvement within the manuscript that the author may want to address.

i) Revised the title. One option could be "Perceived Knowledge, Attitudes and Practices (KAP), and Fear towards COVID-19 among Patients .....".

ii) Some sentences are very short. Some sentences seems to be incomplete. For example, on page 1, line 8-9, "This will help evaluate the gaps ... and provide the appropriate". There should be something mentioned after the gaps. I think you are referring to the gaps in knowledge, attitudes and practices. This is just an example and there are many such places requiring attention.

iii) Revise the write ups addressing typos and improving the grammar. For example, on page 1, line 37, the "End" should be "end".  This is just an example and there are many such places requiring attention. There are some inconsistencies in using upper and lower case letters in Table 2. For example, female should be "Female" and "widow" should be "Widow". These are just an example and there are many such places requiring attention in other sections of the manuscript.

iv) Provide citations for some facts or arguments, if available. For example, for the "The patients’ overall knowledge was categorized as good, moderate, or poor if the total score was between 80 - 100%, 60 - 79%, or less than 60%, respectively." There are a few other places where you would require to provide citations/sources. Please use your intuition as I could not list them in here.

v) It was surprising not to find variation in perceptions on KAP and fear by respondents' demographic characteristics except by employment status, the highest education and marital status. You may want to discuss more why this is the case.

v) Strengthen the discussion section. You may want to avoid repeatedly mentioning the results in the discussion section unless that is essential.

vi) You may want to move the "Implications for Future and Practices" section after or merge it with the "Conclusion" as the flow of any paper warrants the conclusion before the recommendations or implications.

Once again, overall the manuscript is very scholarly and well presented.  

Good luck!

Author Response

Reviewer 1:

Overall, this manuscript is well-written. Congratulations!

However, there are areas for improvement within the manuscript that the author may want to address.

Thank you for your time and effort in revising the manuscript and providing valuable comments. All the comments are addressed point by point (please note that amendments on the manuscript were done in red font)

  1. i) Revised the title. One option could be "Perceived Knowledge, Attitudes and Practices (KAP), and Fear towards COVID-19 among Patients .....".

As advised, the title has been changed to the following: “Perceived Knowledge, Attitude, and Practices (KAP), and Fear towards COVID-19 among Patients with Diabetes Attending Primary Healthcare Centers (PHC) in Kuwait”

  1. ii) Some sentences are very short. Some sentences seems to be incomplete. For example, on page 1, line 8-9, "This will help evaluate the gaps ... and provide the appropriate". There should be something mentioned after the gaps. I think you are referring to the gaps in knowledge, attitudes and practices. This is just an example and there are many such places requiring attention.

I’m unsure whether the manuscript draft was downloaded. Taking the example that was provided by the reviewer, the sentence was complete in the original manuscript as follows:

“This will help to evaluate the gaps and provide the appropriate support to limit the spread of COVID-19 infection across high-risk patients.”

iii) Revise the write ups addressing typos and improving the grammar. For example, on page 1, line 37, the "End" should be "end".  This is just an example and there are many such places requiring attention. There are some inconsistencies in using upper and lower case letters in Table 2. For example, female should be "Female" and "widow" should be "Widow". These are just an example and there are many such places requiring attention in other sections of the manuscript.

Thank you for your comment. The mentioned points were corrected and the whole manuscript was revised for similar editing issues or grammar:

  • Line 37: the space before “the World Health Organization” was deleted
  • Line 59: the space before “Currently, in the absence of an effective….” was deleted.
  • Line 123: the space before “is a validated instrument developed by…” was deleted.
  • Line 151: the space before “Binary logistic regression…” was deleted.
  • Table 2: “Median 52 years (min. 18, max. 87)” was brought in one line. The table was improved in terms of editing.
  • Table 3: The table was improved in terms of editing.
  • Table 4: in the table foot, the word “Two” was re-written in lower case “two”.
  • Table 6: “attitude question” was re-written as “attitude questions”. “n” was reinserted before the “%”
  • Table 7: in the table foot, “Data missing from..” was rewritten as “Data was missing from..”
  • Figure 1: Capitals and small letters were made consistent in the figure legend.
  • Table 8: Editing issues such as capitals and small letters were made consistent. For example, “Reference group” was changed to “reference group”. Details were added in the table foot in reference to the * and ** mentioned in the table.
  • Line 260: extra space before “Kuwait where two third of ….” was deleted.

  1. iv) Provide citations for some facts or arguments, if available. For example, for the "The patients’ overall knowledge was categorized as good, moderate, or poor if the total score was between 80 - 100%, 60 - 79%, or less than 60%, respectively." There are a few other places where you would require to provide citations/sources. Please use your intuition as I could not list them in here.

The questions and method of analysis used in the survey instrument were an adaptation of the measures developed in other similar studies. Taking this point into consideration, the following sentence with citation was included (please see manuscript page 3, lines 102, 109, 118):

Using measures and the cut-off points from other studies (30, 35,36),……..” 

  1. v) It was surprising not to find variation in perceptions on KAP and fear by respondents' demographic characteristics except by employment status, the highest education and marital status. You may want to discuss more why this is the case.

Thank you for this comment. As advised more elaboration was added in the discussion (please refer to page 13, lines 325-340) and the reference list was updated by adding a new ref (no 55):

Unsurprisingly, this study showed that people of postgraduate level had less fear of COVID-19 compared to others. This could be because the level of education positively impacted the psychological well-being of people (53, 54). Similar to the results from the study by Doshi et al, 2021 (55), a significant association was demonstrated between fear and marital status, that is divorced patients had less fear compared to others. This could be attributed to the fact that married people have a greater sense of responsibility and financial commitments towards loved ones compared to others, and the pandemic has affected the security of jobs worldwide. Moreover, unemployed patients in this study had more knowledge about COVID-19 compared to others. This could be because they have more free time than those who are still working; thus, they have greater knowledge about pandemics with better coping skills for emergencies compared to others (56). Non-Kuwaiti patients showed significantly better prevention practices in the current study compared to the Kuwaitis. This could be explained by the healthcare delivery is supported by the government and is free of charge for the citizens. On the other hand, non-Kuwaitis must pay for their medical expenses, which could have affected their adherence to the preventive practices during the COVID-19 pandemic.

  1. v) Strengthen the discussion section. You may want to avoid repeatedly mentioning the results in the discussion section unless that is essential.

As advised, some of the results mentioned in the discussion section were removed and reworded:

  • Page 12, Lines 272-274: The sentence “Most patients in the current study knew the common transmission modes of COVID-19 such as respiratory droplets (97.2%) and touching contaminated surfaces (94.8%), however, only (37.3%) of them knew that the infection does not spread through insulin injections.” was changed to: “Most patients in the current study knew the common transmission modes of COVID-19 such as respiratory droplets and touching contaminated surfaces, however, only a minority knew that the infection does not spread through insulin injections”.

  • Page 12, Lines 283-285: The sentence “In terms of COVID-19 symptoms, 92.8% of patients in the current study identified the main symptoms of COVID-19, but 58.2% could not identify the uncommon ones. In addition, 83.7% of patients were not able to distinguish the symptoms of COVID-19 from those of the common cold.” was changed to “In terms of COVID-19 symptoms, regardless that the majority of patients in the current study were able to identify the main symptoms of COVID-19, most of them failed to distinguish the symptoms of COVID-19 from those of the common cold.”

  • Page 12, Lines 291-293: the sentence “ Regarding knowledge questions on the prevention and control of COVID-19, most patients answered this section correctly and knew that hand washing (94.0%), wearing face masks (94.0%), avoiding crowded places (82.1%), and isolation of infected patients (96.0%) are effective preventive measures.” was changed to: “ Regarding knowledge questions on the prevention and control of COVID-19, most patients answered this section correctly and knew that hand washing, wearing face masks, avoiding crowded places, and isolation of infected patients are effective preventive measures.”

  • Page 12, Lines 300-303: The sentence: “Regarding the attitude, most patients showed a positive attitude (90.4%), in agreement with other studies (40, 41, 43, 44). About 88% of the patients reported that they trust the preventative measures implemented by the MOH, and 52.2% of them strongly agreed that the pandemic will eventually be brought under control; similar to the results from a….” was changed to: “Regarding the attitude, most patients showed a positive attitude, in agreement with other studies (45, 46, 48, 49). Most patients reported that they trust the preventative measures implemented by the MOH, and more than half of them strongly agreed that the pandemic will eventually be brought under control; similar to the results from a …”

  1. vi) You may want to move the "Implications for Future and Practices" section after or merge it with the "Conclusion" as the flow of any paper warrants the conclusion before the recommendations or implications.

As advised this section was moved after the conclusion.

Once again, overall the manuscript is very scholarly and well presented.  

Good luck!

Reviewer 2 Report

The paper investigated the KAP levels of COVID-19 among diabetic patients in selected areas of Kuwait by means of a questionnaire. The results of the 251 questionnaires showed that these diabetic patients had moderate knowledge of COVID-19. This suggests that different strategies are still needed to improve knowledge and support mental health. The study has a good practical value and is helpful for patients to understand COVID-19, but there are still some problems, which will be perfect for publication with the following modifications.

1. the key words of the paper suggest adding COVID-19, which relates to the topic of the paper.

2. Do objective factors such as education level, age, etc., have an effect on KAP? It is recommended to explain in detail.

3. Include a description of the structure of the paper at the end of the introduction, so that the reader can clearly find the part he needs.

4. In the conclusion part of the paper, you should use get data to prove your point.

Author Response

Reviewer 2:

The paper investigated the KAP levels of COVID-19 among diabetic patients in selected areas of Kuwait by means of a questionnaire. The results of the 251 questionnaires showed that these diabetic patients had moderate knowledge of COVID-19. This suggests that different strategies are still needed to improve knowledge and support mental health. The study has a good practical value and is helpful for patients to understand COVID-19, but there are still some problems, which will be perfect for publication with the following modifications.

Thank you for your time and effort in revising the manuscript and providing valuable comments. All the comments are addressed point by point (please note that amendments on the manuscript were done in red font)

  1. the key words of the paper suggest adding COVID-19, which relates to the topic of the paper.

As advised, the word “COVID-19” was added to the keyword. Please see page 1 line 29.

  1. Do objective factors such as education level, age, etc., have an effect on KAP? It is recommended to explain in detail.

Table 8 of the results tested the associations between KAP and demographics (page 11 ) and the results section of the manuscript (page 10, lines 246-255) explained the table:

Binary logistic regression analysis identified that divorced patients and those with postgraduate education were less likely to have perceived fear from COVID-19 (p≤0.05). Furthermore, unemployed and non-Kuwaiti patients were more likely to be knowledgeable about COVID-19 and have better prevention practices, respectively compared to others. On the other hand, there was no statistically significant difference between patients in terms of their attitude toward COVID-19 (p≥0.05). Non-Kuwaiti patients showed significantly better prevention practices compared to Kuwaitis (Table 8).

Other demographic variables were not significantly associated with KAP.

  1. Include a description of the structure of the paper at the end of the introduction, so that the reader can clearly find the part he needs.

As advised the following sentence was added at the end of the introduction (page 2, lines 78-79):

“In the following sections, the methods used to conduct the study, the main results and the discussion will be provided in detail.”

  1. In the conclusion part of the paper, you should use get data to prove your point.

I’m not sure what the reviewer means about “get Data”!

Reviewer 3 Report

Review of the article: “Evaluation of Perceived Fear and Knowledge, Attitude, and  Prevention Practices (KAP) towards COVID-19 among Patients  with Diabetes Mellitus (DM) Attending Primary Healthcare Centers (PHC) in Kuwait”

The scientific problem is important from the theoretical perspective and practice. The Authors use the proper academic terminology. Significant scientific issues are discussed in the article.

The methological rigor of the article is at sufficient level. In order to improve the article, the following should be answered or better discribed: 

-      novelty and motivation for writing the paper;

-      theoretical/empirical research gap;

-      the question of whether other authors have done systematic review in this area of research is worth answering.

Author Response

Reviewer 3:

The scientific problem is important from the theoretical perspective and practice. The Authors use the proper academic terminology. Significant scientific issues are discussed in the article.

The methodological rigor of the article is at a sufficient level. In order to improve the article, the following should be answered or better described: 

Thank you for your time and effort in revising the manuscript and providing valuable comments.

-      novelty and motivation for writing the paper;

-      theoretical/empirical research gap;

      The first two points of the reviewer’s comments are related and were already mentioned in the body of the manuscript. For example regarding the research gap, It was written at the end of the introduction the gap of research in Kuwait with respect to KAP and COVID-19” (please see manuscript page 2, lines 72-74):  “To the best of our knowledge, there is scarce data assessing KAP toward COVID-19 among high-risk patients (28-31) and assessing their perceived level of fear (32-34).”

Regarding the novelty, in the discussion section, it was mentioned that this study was the first in Kuwait to identify the fear of COVID-19 among patients with diabetes and to assess their KAP level (please see the manuscript, page 12, lines 260-261): “To the best of our knowledge, this is the first study to evaluate the fear of COVID-19 among patients with DM attending PHCs in Kuwait and to assess their KAP level.”

This issue was also highlighted as one of the study’s strengths (please see the manuscript, page 13, line 344).

-      the question of whether other authors have done systematic review in this area of research is worth answering.

Thank you for this great point. The following sentence was added to the introduction and references across the manuscript were updated accordingly (Please see the manuscript, page 2, lines 72-73):

“Many systematic reviews and meta-analyses were conducted to identify studies assessing KAP towards COVID-19 across different countries (23-25) and populations (26-27).”